# Transcriptomic Analysis of Light-Induced Genes in *Nasonia vitripennis*: Possible Implications for Circadian Light Entrainment Pathways

**DOI:** 10.3390/biology12091215

**Published:** 2023-09-06

**Authors:** Yifan Wang, Leo W. Beukeboom, Bregje Wertheim, Roelof A. Hut

**Affiliations:** Groningen Institute for Evolutionary Life Sciences, University of Groningen, 9712 CP Groningen, The Netherlands; l.w.beukeboom@rug.nl (L.W.B.); r.a.hut@rug.nl (R.A.H.)

**Keywords:** biological clock, light sensitivity, photoreceptor, transduction pathway, RNAseq

## Abstract

**Simple Summary:**

Almost all animals on Earth, which are subject to daily environmental changes, have evolved an endogenous circadian system to maintain daily timekeeping with their environments. Light serves as the most crucial external cue for pollinating insects, such as bees and wasps, enabling them to navigate, forage, and reproduce. It is essential to understand the impact of light on circadian systems in animals, especially considering the prevalence of artificial light nowadays. In our study, we aimed to study the possible molecular mechanisms of the circadian light input pathway in the jewel wasp, *Nasonia vitripennis*, by utilizing a transcriptomic sequencing approach. We proposed a model for such a light input pathway that involves multiple photoreceptors, neurotransmissions, and light-induced core clock gene expressions. Our study represents the initial step in understanding the circadian light input pathway in wasps and provides interesting candidates for future testing.

**Abstract:**

Circadian entrainment to the environmental day–night cycle is essential for the optimal use of environmental resources. In insects, opsin-based photoreception in the compound eye and ocelli and CRYPTOCHROME1 (CRY1) in circadian clock neurons are thought to be involved in sensing photic information, but the genetic regulation of circadian light entrainment in species without light-sensitive CRY1 remains unclear. To elucidate a possible CRY1-independent light transduction cascade, we analyzed light-induced gene expression through RNA-sequencing in *Nasonia vitripennis*. Entrained wasps were subjected to a light pulse in the subjective night to reset the circadian clock, and light-induced changes in gene expression were characterized at four different time points in wasp heads. We used co-expression, functional annotation, and transcription factor binding motif analyses to gain insight into the molecular pathways in response to acute light stimulus and to form hypotheses about the circadian light-resetting pathway. Maximal gene induction was found after 2 h of light stimulation (1432 genes), and this included the opsin gene *opblue* and the core clock genes *cry2* and *npas2*. Pathway and cluster analyses revealed light activation of glutamatergic and GABA-ergic neurotransmission, including CREB and AP-1 transcription pathway signaling. This suggests that circadian photic entrainment in *Nasonia* may require pathways that are similar to those in mammals. We propose a model for hymenopteran circadian light-resetting that involves opsin-based photoreception, glutamatergic neurotransmission, and gene induction of *cry2* and *npas2* to reset the circadian clock.

## 1. Introduction

Almost all organisms on our rotating Earth are subject to daily and seasonal variations in the environment, such as variations in light intensity, photoperiod, and temperature. Organisms have evolved endogenous circadian and seasonal rhythms to anticipate these environmental changes and optimally capitalize on environmental resources [1]. For example, circadian systems provide an internal time reference for foraging, for sun compass navigation and time–place-learning, and for reproduction or hibernation regulation by measuring changes in day length [2]. Light is the most important and predictable cue that animals use to keep in synchrony or entrain to the 24 h daily environmental changes [3]. Disruption of the internal clock has strong negative effects on fitness. In humans, for example, adverse health implications for sleep function and mood have been reported following light disruption and desynchronization [4]. In animals, including insects, circadian disruption of the clock can negatively affect the immune system, reproductive regulation, and metabolic functions [5,6,7,8,9].

The circadian clock system can be influenced by light through different mechanisms. For instance, the circadian clock can be entrained by specific light–dark cycles or can be shifted by abrupt light pulses, resulting in clock resetting. To understand the impact of environmental light on the circadian system, it is crucial to understand the molecular mechanisms underlying different circadian light input pathways. So far, the paradigm for *Drosophila* is that light information can be received directly by a light-sensitive clock component, CRYPTOCHROME (dCRY, CRY1 type), and by opsin photopigments in compound eyes, Hofbauer–Buchner eyelets, Bolwig’s organs, and ocelli. Light activation of *Drosophila*-like CRY can subsequently reset the circadian clock by inducing the degradation of TIMELESS (TIM) [10]. In contrast, the paradigm for mammalians is that light input is conveyed to the clock exclusively through the retina, involving melanopsin as the neuronal photoreceptor, which is expressed in intrinsically photosensitive retinal ganglion cells (ipRGCs), and rod and cone opsins are expressed in the retinas [11,12,13,14,15]. Signals from these cells are then transmitted to the suprachiasmatic nucleus (SCN, the circadian pacemaker) and reset the circadian clock by inducing gene expression of the clock gene *period* (*per1* and *per2*) through a cascade transduction [16,17,18]. In other non-mammalian vertebrates, especially fish, light information can be directly detected by cells and tissues to entrain the local clocks [19,20,21]. Downstream of the photoreceptors and their signaling pathways, light triggers the acute induction of the core clock genes *cry1a* and *per2* [20,22,23]. Taken together, these examples illustrate two different types of circadian light-resetting mechanisms: through either protein level regulation (degradation), as in *Drosophila*, or light-induced transcriptional change, as in vertebrates [24]. 

Although the circadian system is evolutionarily conserved across a wide range of species, there is tremendous diversity in the compositions of the clock systems of insect species [25,26]. Light transduction pathways remain largely unknown in insect species beyond *Drosophila*. Interestingly, light-sensitive CRY1 has been lost multiple times in evolution. Some insects, such as Hymenoptera (sawflies, bees, wasps, and ants), Blattaria (cockroaches), and Coleoptera (beetles), possess only light-insensitive CRY2 [27,28], whereas other insects, such as Lepidoptera (butterflies and moths) and Orthoptera (grasshoppers and crickets), possess both types of CRY [26]. The clock system of hymenopterans includes *mcry* and *timeout* as the core clock genes, as well as the *mper* (*mammalian-like period*), *clock* (*circadian locomotor output cycles kaput*), *npas* (*neuronal pas domain protein*), and *cycle* genes [29]. In fact, molecular and phylogenetic analyses revealed that the hymenopteran clockwork has more similarities with the mammalian clockwork than with that of *Drosophila* [30]. Transcriptional profiles also indicate little similarity between cycling genes in *Drosophila* and hymenopteran insects [31]. Therefore, we hypothesized that the mechanisms underlying the circadian light input pathway in Hymenoptera may be similar to those in mammals and other vertebrates, involving light-induced induction of circadian genes and circadian regulation of transcription. To better understand how light information is processed and transmitted to the central clock in Hymenoptera, analyzing gene transcriptional changes in response to light can provide essential insights into this process. 

Previous research has demonstrated that *Nasonia vitripennis* is a suitable research model for hymenopteran insects [32]. *N. vitripennis* is a small parasitoid wasp that is easy to rear, and its genome is fully sequenced [33]. *Nasonia*’s circadian properties are well characterized: It has robust circadian responses in locomotor activity and precise timing of adult emergence [34,35], a strong light-resetting mechanism [36], and robust photoperiod response in diapause induction [37,38]. To identify the circadian-controlled cycling transcripts in *Nasonia*, the circadian transcriptome of *Nasonia* under full darkness (DD) and full light (LL) conditions was investigated [31]. To identify how the light input pathway regulates circadian controlled expression, we can manipulate light to disrupt, reset, or stimulate the circadian clock. For example, abrupt light stimulation can induce expression changes in the core clock gene, thus regulating a cascade of downstream reactions and shifting the animal’s activity patterns. The magnitude of the shift in the animal’s activity pattern increases with the increased duration of light pulses [39]. By entraining *Nasonia* with different monochromatic light pulses, our previous research indicated that multiple opsins in the compound eye and ocelli of *Nasonia* are involved in the regulation of circadian light resetting, with blue-, green-, and red-sensitive opsin photoreceptor cells being the major photoreceptors [40]. This is consistent with the roles of opsin photoreceptors and photoreceptive organs in circadian entrainment in cockroaches and crickets [41,42,43]. 

In this study, we used RNA sequencing (RNA-seq) on wasp heads to profile light-stimulated transcriptome gene expression at several time points during 4 h of light exposure. We analyzed the effects of acute light stimulation on entrained wasps. We aimed to deduce a hypothetical sequence of actions relevant to the light transduction cascade underlying entrainment by light pulses. We aimed to elucidate if such a cascade leads to the transcriptional activation of the core clock genes in *N. vitripennis*. Co-expression analysis was performed to identify modules of genes with covarying expression patterns and search for overrepresented functional annotations for these gene modules. We discuss our results in comparison to the gene expression patterns of light induction in other animals. 

## 2. Materials and Methods

### 2.1. Experimental Lines and Maintenance

The standard *N. vitripennis* laboratory strain AsymC was used [33,44,45]. *Nasonia* cultures were maintained in a temperature- and humidity-controlled incubator (20 ± 1 °C, 50–55% RH) under a light–dark cycle of 18:6 LD (645.83 lux) to prevent diapause induction. All wasps were kept in plastic vials (70 × 20 mm) with the pupae of *Calliphora* spp. flies as parasitic hosts for egg laying and feeding. About 20–30 wasps with 30–50 fly pupae were maintained in each generation, with a generation time of approximately 21 days at 20 °C. Previous studies showed that male wasps are extremely short-lived and have less rhythmicity than female wasps [35,46]. Therefore, only female wasps were used in this circadian experiment. To obtain wasps for the experiments, 1–2-day-old mated females were individually housed on 2–3 fly pupae in cotton-plugged polystyrene tubes (60 × 10 mm) to avoid harmful effects of crowding on longevity.

### 2.2. Entrainment and Sample Collection

For all experiments, females at the black pupal stage (approximately 17–18 days after egg laying) were collected from the fly pupae and placed in cotton-plugged polystyrene tubes with sugar water strips (10 wasps in each tube). After eclosion, females (1–2 days old, unmated) were transferred into cotton-plugged polystyrene tubes filled to one-quarter with agar–sugar food (30% sucrose, 1.5% agar, 0.1% nipagin). All female wasps were entrained prior to sample collection using the following protocol: Tubes with wasps were placed horizontally at the bottom of light-tight boxes (23 × 14 × 32 cm) to receive an equal amount of light in a temperature- and humidity-controlled climate room (18 ± 1 °C, 50–55% RH). The light source in each light-tight box was one LED light (ILH-GD01-NUWH-SC201, Neutral White 4000 K, PowerStar, Berkshire, UK), which illuminated the wasps with broadband white light of approximately 2.10 × 10^15^ photons·cm^−2^·s^−1^ after passing through a light diffuser sheet. 

In our previous research, we optimized the entrainment protocol for inducing the maximal phase-shift responses in *Nasonia* [40]. To further investigate the molecular pathway underlying the circadian light entrainment pathway, the same protocol as that described before was used [40]. All wasps were initially entrained under a light–dark cycle of 14:10 LD for 7 consecutive days. The LD cycle was switched off on day 8, and a light pulse was given in the middle of their subjective night at ZT18 after 28 h of darkness. After this, wasps were sampled at four time points—0.5 h, 1 h, 2 h, and 4 h after the onset of a continuous light treatment. Wasps were thus subjected to various lengths of light pulse regimes (Figure 1) to determine the dynamics of light-stimulated gene induction, which will potentially provide further clues as to which mechanisms underlie circadian light resetting. As a control treatment, female wasps were entrained under a light–dark cycle of 14:10 LD for 7 consecutive days and were left in separate light boxes in constant darkness without receiving a light pulse. The control wasps were sampled at the same four time points. All room light sources were removed from the light-sealed climate-controlled room, and control samples were collected in darkness without any additional environmental light influence. For both light treatment and dark control conditions, wasps were collected and snap-frozen in liquid nitrogen at ZT18.5 (0.5 h of treatment), ZT19 (1 h of treatment), ZT20 (2 h of treatment), and ZT22 (4 h of treatment), following the scheme in Figure 1. Three biological replicates (each consisting of 100 female wasps) were sampled per time point for the light exposure, and two biological replicates (each consisting of 100 female wasps) were sampled per time point for the dark controls. All samples were stored in a −80 °C freezer until RNA extraction.

### 2.3. RNA Isolation and RNA Sequencing

We performed RNA sequencing of the wasp heads to characterize the light transduction cascade from the photoreceptors in the compound eye and ocelli to the circadian clock in the brain. After snap freezing, wasp heads (100 per sample) were carefully removed with a sharp razor under a microscope for RNA extraction. Total RNA was isolated using the TRIzol^®^ reagent (Thermo Fisher Scientific, Invitrogen, Carlsbad, CA, USA) protocol in an RNase-free environment. Head tissues were suspended in TRIzol^®^ and homogenized with a 5 mm RNase-free stainless-steel bead in a TissueLyser II machine. Chloroform was used to separate RNA, DNA, and proteins; the upper aqueous phase was carefully collected and further treated with isopropanol at room temperature, and the RNA pellet was precipitated by centrifugation at 4 °C. The RNA pellet was further washed 3 times in 75% RNase-free ethanol and redissolved in RNase-free water. The concentration and purity (A260/A280 and A260/A230 ratio) of the RNA samples were measured with Nanodrop 2000 c (Thermo Scientific, DE). All RNA samples were then sealed, packed on dry ice, and shipped to Novogene (Cambridge, UK) for RNA quality assessment, library preparation, and sequencing.

The quantitation, integrity, and purity of all RNA samples were checked at Novogene with Nanodrop, an Agilent 2100 Bioanalyzer, and agarose gel electrophoresis. All samples were of high integrity based on the Bioanalyzer profiles and agarose gel, and 1 μg of each sample RNA was then used for cDNA library preparation and RNA sequencing. Each sample was 150 bp paired-end sequenced on the Illumina Novaseq6000 platform, providing >30 M raw reads per sample, with >80% of the reads having at least a Q30 quality score. The raw sequencing reads presented in this study were deposited in the European Nucleotide Archive (ENA) under accession no. PRJEB57723.

### 2.4. Differential Expression and Functional Annotation Analysis of RNA-Seq

RNA-seq analysis was performed by following the “new Tuxedo” protocol [47] in R v4.0.3 [48] on the Peregrine and Millipede computer clusters of the University of Groningen. Firstly, the quality of the sequencing data was checked with FastQC v0.11.7 [49], MultiQC v1.0 [50], and FastQ Screen v0.14.1 for contamination [51]. Adapters and low-quality reads (Q < 30) at each end (leading and trailing) were trimmed off with Trimmomatic v0.39 [52]. 

The clean data were mapped to the newest NCBI *Nasonia* reference genome “GCF_009193385.2_Nvit_psr_1.1_genomic.fna” [http://ftp.ncbi.nlm.nih.gov/ (accessed on 3 May 2021)) using HISAT2 v2.2.1 [53]. For a mapping quality check, Picard tools v2.18.17 (Picard Tools—By Broad Institute), including MarkDuplicates, CollectAlignmentSummaryMetrics, and CollectRnaSeqMetrics, were used, and optical duplicates caused by sequencing errors were removed from the sequencing data, thus achieving a read mapping ratio of 95–97% in the end. Next, the alignments were passed to StringTie v2.1.4 [54] for the assembly and quantification of the expressed genes and transcripts. Additionally, gffcompare v0.11.2 [55] was used to compare the StringTie output GTF files with the reference annotation file again to identify the predicted transcripts related to the annotation file. All predicted transcripts were extracted from the merged GTF file from StringTie and blasted against the NCBI RefSeq non-redundant (nr) database for further verification using Diamond [56]. Novel transcripts without a homolog identification in the nr database were further removed from the merged GTF file from StringTie and then fed back to StringTie to generate read count data for statistical analysis. For more functional annotation, the assembled transcripts were aligned to the nr, UniProtKB/Swiss-Prot, and UniProtKB/TrEMBL databases with Diamond (e-value ≤ 1 × 10^−5^). Gene Ontology (GO) terms and Kyoto Encyclopedia of Genes and Genomes (KEGG) terms were retrieved from the UniProt and KEGG pathway databases, and they were additionally searched using InterProScan [57] against the Superfamily [58] and Pfam [59] protein databases.

Gene expression levels were statistically analyzed using DESeq2 [60] in R, and they were grouped according to treatment, treatment duration, and batch conditions. Count data were filtered and supplied to the DESeq2 model, where the model performed estimation of size factors, estimation of dispersion, and negative binomial GLM model fitting. Principal component analysis (PCA) was used to reduce the dimensions (Appendix A). No clear correlation was found between PC1 and PC2 for either treatment or treatment duration, which could be because not many genes responded to light. One of the biological replicates for light-1h deviated somewhat from most other samples, which may have resulted in lower power in detecting differences in gene expression for that time point. All samples were included in the further analysis. In our study, we were particularly interested in light-induced changes in gene expression profiles. Therefore, we compared the expression levels of light treatment (LT) and dark control (DC) at each respective time point, i.e., LT 0.5 h vs. DC 0.5 h, LT 1 h vs. DC 1 h, LT 2 h vs. DC 2 h, and LT 4 h vs. DC 4 h. Wald statistics were in the gene expression comparison at each time point, and the Benjamini–Hochberg algorithm was used to adjust *p*-values. Differentially expressed genes (DEGs) were defined as all genes that were significantly differentially expressed at least once in one of the four compared groups with a false discovery rate (FDR) of <0.05. The hierarchical clustering of global gene expression was performed using pheatmap with the complete-linkage clustering method and Euclidean clustering distance [61]. GO-based overrepresentation analysis was performed for each contrast comparison using TopGO and ViSEAGO [62]. For the genes that were differentially expressed at a time point or for a cluster of DEGs that had a particular expression profile over the 4 timepoints combined, we tested for overrepresentation. GO terms were tested with the Fisher exact test, and the *elim* algorithm was used to adjust *p*-values. GO terms were considered significant at *p* < 0.01. KEGG pathway overrepresentation analysis was performed using ClusterProfiler [63], and the Benjamini–Hochberg algorithm was used to adjust *p*-values. KEGG pathways were considered significant at *p* < 0.05.

Additionally, in our study, we were interested in the light-stimulated expression pattern changes over the time course. A co-expression analysis of the expression changes of the differentially expressed genes over time (the four time points) was conducted using TCseq [64]. This revealed that the expression pattern of the DC samples also changed over the time course. To account for the changes in expression over the course of the 4 h in darkness, a co-expression analysis was performed on the differences in relative expression between the light and dark treatments over the time course—thus, the expression in darkness was extracted from the expression under light conditions at the same time point. Subsequently, GO-based overrepresentation analysis was also performed on those modules. Motif enrichment analysis (MEA) was performed to determine whether there were common DNA-binding transcription factors that controlled the transcription of each module of genes by detecting the enrichment of known binding motifs in the regulatory regions of those genes. SeqKit toolkit [65] was used to retrieve the 500 bp upstream flanking sequences of all DEGs. Duplicated sequences and sequences shorter than 8 bp were removed. HOMER [66] was then used to search for known overrepresented motifs in each cluster of genes with the default findMotifs setting. The enriched motifs were detected by searching against known motifs in *Drosophila* and mammal databases in HOMER (v4.11). However, not much is known about the transcription factor binding motifs in *Nasonia*, which implies that these should be considered as “putative” motifs.

## 3. Results

### 3.1. Differentially Expressed Genes Induced by Light

The RNA-seq analysis generated, on average, 32,681,629 raw reads per sample and 31,260,985 clean reads after quality control (electronic Appendix A). Approximately 95% of the reads, corresponding to 15,904 genes, were successfully mapped to the *Nasonia* reference genome. Sixteen percent of the reads were mapped to novel exons, introns, and loci (electronic Appendix A).

After examining the gene expression profiles of the wasp heads in the dark, it was noticed that the gene expressions changed over the 4 h time course. Moreover, we expected dynamic expression for genes that regulated the response to light stimulation, with many genes showing only changes in expression for a subset of the time points. Therefore, to study the effect of light on the gene expression profiles of the wasp heads, we compared the gene expression levels of the light-stimulated samples relative to the expression in DC at each time point. We found 277, 381, 1432, and 147 genes to be differentially expressed after 0.5 h, 1 h, 2 h, and 4 h of light treatment, respectively, at an FDR threshold of 0.05 (Figure 2). The number of genes that were influenced by light—either upregulated or downregulated—increased with the duration of light treatment and peaked after 2 h of light treatment. Overall, we defined 1886 genes as differentially expressed genes (DEGs) that showed significant differences in expression between LC and DC for at least one of the time points; a summary of the genes with significant expression changes after light stimulation at any time point is provided in the electronic Appendix A. 

### 3.2. Functional Annotation Analysis of Differentially Expressed Genes

To further characterize the expression profiles, we first visualized the global expression of all 1886 DEGs in all samples (Figure 3a). A hierarchical clustering analysis showed that these DEGs exhibited various expression patterns over time (Figure 3a). One large cluster of genes (cluster 1) was downregulated from 0.5–2 h after the onset of light exposure compared to the dark controls. Another large cluster of genes (cluster 2) was generally upregulated, especially at 0.5 h and 2 h, but with some variations at 1 h and 4 h. A smaller cluster of genes (cluster 3) showed more variable expression among the biological replicates, as it was upregulated or downregulated in a subset of samples or time points. 

To explore the biological significance of the genes that were differentially expressed at the four time points, we used the TopGO bioinformatics tool to perform Gene Ontology (GO) annotation analysis on upregulated and downregulated genes at each time point. Out of the 1886 DEGs, 1091 had GO terms, which reflected the limitations of GO annotation for *Nasonia*. Using overrepresentation GO enrichment analysis with an adjusted *p*-value threshold of 0.01, the DEGs were classified into 77 enriched functional groups consisting of 44 biological processes, 20 molecular functions, and 13 cellular components (electronic Appendix A). The GO analysis showed that the largest numbers of DEGs belonged to the categories of metabolic and cellular processes in biological processes, oxidoreductase activity in molecular functions, and ribosome and cytosol in cellular components (Figure 3b,c). 

Notably, within 0.5 h after light exposure, light triggered the enrichment of basic metabolic process, fatty acid biosynthesis, and enzyme activities, lasting until at least 2 h after the light exposure. After 0.5 h of light exposure, the downregulated DEGs were enriched in nucleosome, chromatin assembly, and rRNA synthesis (rRNA maturation and preribosome), indicating the loosening of the chromatin structure and light-triggered cellular processes (Figure 3b,c). There was an increased number of DEGs at 1 h, with most of the upregulated DEGs being associated with protein secretion and transport, the glutamate metabolic process, meiosis, mitosis, and apoptosis. This indicated that 1 h after light exposure, there were active light-induced cellular division processes (Figure 3b,c). The highest number of DEGs occurred at 2 h (1432 DEGs), consisting mainly of those involved in histone modification, ribosomes, translation, the neurotransmitter process, and GABA receptor activity (Figure 3b,c). In contrast, only 147 genes had a differential expression at 4 h, and they were mostly involved in membrane transport and signaling, including the enrichment of transmembrane transport, the intrinsic component of membrane, and the response to lipids and organic cyclic compounds (Figure 3b,c).

Taking different sets of enriched GO terms together, the majority of the enriched GO terms were time-point-specific. Out of the 77 enriched functional groups, only palmitoyltransferase activity was upregulated in the first three time points, and multiple Coenzyme A synthase activities were upregulated after both 0.5 h and 2 h of light exposure (Appendix A). On the other hand, out of the 1886 DEGs, only 48 had Kyoto Encyclopedia of Genes and Genomes (KEGG) annotations. This limited the power of KEGG pathway analysis in *Nasonia*. Nonetheless, the KEGG analysis revealed similar enriched gene pathways to those of the GO analysis: the enrichment of basic metabolism, fatty acid biosynthesis, and ribosomes (Figure 4; electronic Appendix A).

To piece together a possible cascade of phototransduction related to the circadian light-resetting pathway, we were particularly interested in the sequences of actions of genes over time. We performed a time-course co-expression analysis that clustered all of the DEGs based on their temporal expression patterns. There were also fluctuations in the gene expression in the control samples. This could be the result of cyclic expression changes in the dark phase in those entrained animals. Therefore, relative expression was used in the analysis to control for the changes in expression in the DC samples over the time course. Time-course clustering grouped these 1886 DEGs into ten clusters (Figure 5; electronic Appendix A). Cluster 1 and cluster 6 showed an immediate light effect on the gene expression level (the highest and lowest expression at 0.5 h, respectively), whereas clusters 5 and 10 showed a delayed light effect (the highest and lowest expression at 4 h, respectively), and the other clusters showed peaks or dips in relative expression at intermediate time points. These time-course clusters reflect the sequence of action for differentially regulated but co-expressed genes. 

We analyzed the enriched GO pathways in each time-course cluster (Figure 5), revealing light-induced metabolic, enzymatic, and cellular processes, along with detailed cell division, gene regulation, and protein processing (Figure 5; Appendix A). Cluster 1 (Figure 5a, peak expression at 0.5 h) lacked enriched GO annotations, while cluster 6 (Figure 5f,l, lowest expression at 0.5 h) featured gene silencing, histone modification, and protein binding. Cluster 2 (Figure 5b,k, highest expression at 1 h) showed photoreceptor cell development enrichment, potentially indicating the start of this light input pathway from photoreceptors. Cluster 7 (Figure 5g,l, lowest expression at 1 h) had enriched metabolic and biosynthetic pathways. At 2 h, cluster 4 (Figure 5d,k) showed translation and RNA-binding dominance, whereas cluster 3 (Figure 5c,k) exhibited fatty acid metabolism. Clusters 8 and 9 (Figure 5h,i,k,l) had the lowest expression at 2 h, which was tied to signal transduction, GABA receptor activity, calcium ion transmembrane transport, and organelle processes. The genes that were highly expressed at 4 h (cluster 5 (Figure 5e,k)) were linked to protein processing and peroxisome function. Cluster 10 (Figure 5j,l, lowest expression at 4 h) featured protein binding and fatty acid metabolism. Neuronal activation was indicated in cluster 9 (GABA receptor activity) and cluster 10 (neurotransmitter catabolic process). When evaluating the similarities and differences in the biological processes that were enriched in different clusters, only a few similarities were found. Ribosomes and cytosol were enriched in clusters 3 and 4, whereas several Coenzyme A synthesis activities were enriched in clusters 3 and 10 (Appendix A).

### 3.3. Candidate Genes for the Circadian Light Input Pathway

Although circadian-rhythm-related gene sets and pathways were not significantly enriched in the functional annotation analysis, several DEGs that showed light-induced changes in gene expression were central clock genes or had a putative functional annotation in circadian rhythms or photo-entrainment. To identify the DEGs and pathways that were potentially involved in the regulation of circadian light resetting, we selected several DEGs with relevant functional annotations to look at their expression patterns across the time points and treatments in more detail. 

The *opblue* gene encoding for blue opsin is one of the main circadian photopigments in *Nasonia* [40] and was significantly induced at 2 h; although non-significant, it already appeared to be upregulated at 0.5 h and 1 h (Figure 6). Most interestingly, the central clock genes *cryptochrome 2* (*cry2*) and *neuronal pas domain protein 2* (*npas2*) were also significantly upregulated after 2 h of light treatment compared to the dark control, with *cry2* also already showing higher expression at 0.5 h and 1 h (Figure 6). Additionally, *beta-trcp* was significantly downregulated at 2 h; this is a possible *Nasonia* homolog of the *slimb* gene, which is implicated in PER protein degradation in *Drosophila*. An unknown clock output gene that was annotated as a circadian-clock-controlled protein-like gene was upregulated at 1 h before returning back to the control-level expression at 2 h. The downregulation of *transformer* (*tra*) and the release of juvenile hormone both lead to increased diapause response. Interestingly, *tra* was found to be downregulated at 2 h, while *juvenile hormone acid O-methyltransferase* (*jhamt*) was upregulated at 2 h. In contrast, no significant transcriptomic responses were observed in other opsin genes or core clock genes (Appendix A). 

Light information, which is detected by photoreceptors, is forwarded to downstream neuronal networks, such as the clock neurons, via neurotransmitters. The glutamate pathway was particularly triggered by light stimulation in *Nasonia,* with the expression levels of eight glutamate-related receptor genes being significantly changed (Figure 7). Four of those receptor genes were significantly upregulated after light exposure: *nmdar1-2* (*glutamate [nmda] receptor subunit 1*, LOC116418325) responded to light immediately and was upregulated until 2 h of light treatment; *nadph* (*putative glutamate synthase [nadph]*) was upregulated at 1 h, while *glutamate receptor* (LOC100678247) and *nmdar1-1* (LOC100116847) were upregulated at 2 h (Figure 7). On the other hand, *nmda 2b* (*glutamate receptor ionotropic*) was significantly downregulated at 1 h and 2 h of light treatment, while three other receptors, including *glutamate receptor 1*, *mglur-like* (*metabotropic glutamate receptor-like*), and *nmdar1-like* (LOC103315526), were all downregulated after 2 h of light stimulation (Figure 7).

### 3.4. Candidate Transcription Factors Involved in Circadian Light Responses

Beyond the candidate genes, many transcription factors were detected among the DEGs (electronic Appendix A). It is conceivable that light activated some of these transcription factors, influencing the transcription of co-expressed genes in light response pathways related to visual processing, circadian rhythms, and physiological and behavioral light responses. To delve deeper into the molecular pathways of light responses, we conducted a motif enrichment analysis (MEA) to identify overrepresented transcription factor binding motifs in the regulatory regions of genes within each co-expression cluster. We performed a motif analysis with both insect and mammalian databases to leverage the high conservation of transcription factor binding specificities between insects and mammals [67], as well as the similarities between *Nasonia*’s clock system and that of vertebrates. However, due to limited knowledge about wasp motifs, all putative motifs require subsequent functional testing.

In total, 505 known motifs were found to be enriched in all 10 clusters when using a *p*-value threshold of 0.01 (electronic Appendix A). The enrichment of the known binding motifs for differentially expressed transcription factors is particularly interesting, as it could provide insight into the genetic regulation of the photic response. Coherence between overrepresented motifs and differentially expressed transcription factors included the transcription factors *creb* (*camp response element-binding protein* and two *creb -binding protein-like*), *ap-1* (*activator protein-1 complex subunit beta-1* and *activator protein-1 complex subunit mu-1*), *tbx20* (*T-box transcription factor tbx20-like*), and *znf467* (*zinc finger protein 467*).

Both the CREB/CRE and AP-1 signaling pathways, which have been implicated in the regulation of circadian rhythms in mammals, were differentially expressed, and clusters of genes with corresponding binding motifs changed expression at the same time. The expression levels of several *creb* transcription factors responded to light stimuli. The expression level of *creb* appeared to be higher under light conditions compared to the dark control from the start of light stimulation, and this light-induced difference increased and became significant at 2 h (Figure 8a). One of the transcription coactivators, the *creb-binding protein-like* (*crebbp1*, LOC103315501) gene, was significantly downregulated at 2 h, while the other *crebbp2* (LOC100679057) gene was already significantly downregulated at 1 h. The three overrepresented CREB-binding motifs shared a highly conserved region of TGACGTCA, which is a known mammalian CREB-binding motif (Figure 8b–d). These motifs were found to be enriched in clusters 3, 7, and 10, which represent genes that all showed high expression levels at 2 h of light exposure. Lastly, subunits of the same transcription factor, *ap-1*, were also found to be significantly affected by light treatment, with one subunit (*ap1m1*) being significantly upregulated and another subunit (*ap1b1*) being significantly downregulated at 2 h (Figure 8e). One of the AP-1-binding motifs (Figure 8f) was found to be enriched in clusters 4 and 10, and it shared the conserved sequence TGACTCATC with many known human and mouse motifs in the *fos*-related or *jun*-related transcription factor family. Those motifs were not found in the *cry2* and *npas2* clock genes in the motif search (500 bp upstream). However, an expanded search (2000 bp upstream) revealed that *cry2* possesses the conserved binding site TGACGTC, whereas *npas2* contains both TGACGT and TGACTC.

Additionally, two transcription factors that were implicated in development and cell differentiation were downregulated during the light exposure. The transcription factor *tbx20* was significantly downregulated after 2 h of light exposure (Figure 9a), and, interestingly, a possible TBX20-binding motif was found to be overrepresented in the co-expressed clusters of genes in clusters 4 and 8 (Figure 9b). Both clusters of genes showed a delayed light effect with upregulated expression only 2 h after the onset of the light treatment (Figure 5c,d). This motif presents a highly conserved sequence of GGTGYTGA, as in the known *tbx20* motif in humans (Uniprot ID: Q9UMR3). The downregulated expression of the *tbx20* transcription factor and the concurrent upregulation of two clusters of genes are in line with previous research that showed that *tbx20* can function as both a transcriptional activator and a repressor [68]. T-box transcription factors are implicated as important regulators for development and cell differentiation, including neurogenesis [68,69]. Similarly, the transcription factor *znf467* was significantly downregulated from 0.5–1 h of light treatment before gradually returning to the same expression level as that of the control at 4 h (Figure 9c). This early-onset light effect on *znf467* may correspond with the immediate response in cluster 3, where the enriched ZNF467-binding motif was found (Figure 9d). The functional roles of *znf467* are largely unknown, but it is implicated in cell differentiation in adipose tissue [70].

## 4. Discussion

Our previous research [40] revealed a strong phase-delay effect (approximately 12 h of phase delay) in *Nasonia* when receiving a light stimulation in the middle of the subjective night. However, many aspects of the underlying light-resetting mechanisms are still unknown. Previous research proposed a sequence of events following a phase-shifting light pulse: a brief neural response in seconds and an acute but short-lived molecular response within the first hour, followed by an early phase-resetting response, leading to consolidated expression shifts [71]. To explore this, we applied the same light entrainment and then light stimulation protocols for triggering maximal phase-shift responses and examined the possible consequence of action with the transcriptomic approach. We tested three biological samples under light conditions and two biological replicates under dark conditions per time point. Although transcriptomic analysis with these sample sizes may limit the statistical power, especially for detecting subtle differences in gene expression, it nonetheless offers novel insights and potential candidates for future exploration of the hymenopteran circadian light input pathway. Combining different bioinformatic routes, we searched for possible gene associations with the circadian light-resetting pathway in *Nasonia*. By sampling at four time points after the onset of the light exposure, we could analyze the temporal expression changes and the sequence of action of different genes. Our analysis highlighted the possible temporal sequence of action for many genes that were significantly differentially expressed after light exposure, with certain genes responding in similar patterns. While interpreting these patterns remains challenging due to temporal resolution constraints, the findings lay a foundation for constructing explicit hypotheses.

While the limited sample size affects the statistical power and subtle light-induced expression detection, our RNA-seq experiment did allow us to identify many genes that were associated with the light-transduction cascade and their sequences of action. To detect acute but transient light responses, we opted for a per-time-point analysis instead of combining all data across the four time points. The latter approach may have risked producing false negatives due to the inherent statistical limitations of our unbalanced design and the dynamic expression patterns in the dark control samples. Given that our study aimed to take the first step in generating a hypothesis regarding hymenopteran light-resetting mechanisms, we chose a statistical test with more sensitivity for identifying putative genes involved in the full light-induced cascade. However, this per-time-point analysis might have been more susceptible to false positives. As a subsequent step, the proposed candidate genes require further functional testing through silencing or overexpression in order to confirm their roles in the process of interest. It is important to note that these downstream functional analyses are still relatively limited in *Nasonia* and are beyond the scope of this manuscript. 

In total, the transcription levels of 1886 genes in the head were significantly differentially expressed after receiving light stimulation in the middle of the subjective night compared to the dark control conditions. An immediate light effect was observed after 0.5 h of light on a small set of genes (128 and 149 genes were significantly upregulated and downregulated, respectively). Those genes were predominantly involved in metabolic processes, fatty acid biosynthesis, and enzymatic activity. Similar gene sets were found to be enriched when comparing the profiling of global transcriptional oscillation in *Nasonia* under constant light conditions to those under constant dark conditions [31]. It is possible that those genes may simply have been responding to changes in light in the environment rather than circadian regulation. Indicators of the dynamic remodeling of chromatin were present from 0.5 to 2 h; for instance, there was downregulation of genes with annotations for nucleosomes and chromatin assembly at 0.5 h, as well as upregulation of chromatin binding and condensed chromosomes at 1 h, followed by the downregulation of histone modification at 2 h. This result agrees with earlier work in *Drosophila*, where light-activated chromatin remodeling genes were also shown after 1 h of a light stimulus [72]. This also aligns with the involvement of rapid acetylation and deacetylation in the circadian clock in mammals [73,74,75].

The chromatin changes in *Nasonia* could indicate light-triggered gene replication for cell division, gene expression, or both. An induced light-triggered cell division process was further supported by enriched gene sets, such as those of female meiotic nuclear division, mitotic recombination, chromosome organization involved in meiosis, and meiosis I, at 1 h. On the other hand, light-induced gene expression was also increased by the upregulation of translation, ribosome, and ribonucleoprotein complexes after 2 h of light exposure. Additionally, the number of DEGs increased with the prolonged light stimulation and peaked at 2 h before declining to the lowest number of DEGs at 4 h. Behaviorally, the transition from weak to strong light resetting also occurred after approximately 1 to 2 h of light pulses [39]. These findings suggest that the overall timeframe of light-triggered gene expression occurred within the first 2 h after stimulation. Taken together, the results of the functional annotation analysis revealed a timeline of the enrichment of gene annotations that were involved in the regulation of histone remodeling and chromatin modification in the first hour, gene transcription and translation particularly around the 2nd hour, and protein processing and transmembrane transport from around 1 h until the end of our 4 h experimental window. 

We further characterized the sequences of genes being up- or downregulated in more detail to reveal the cascade of reactions that were triggered throughout the light stimulus. We selectively investigated genes implicated in the circadian literature. For hymenopterans such as *Nasonia*, opsins in their photoreceptive organs are primary candidates for light entrainment, which was confirmed for several insects [41,42,43]. We found evidence of the upregulation of the gene expression of *opblue* opsin by light and the enrichment of photoreceptor cell development as potential first steps for processing light information. The upregulation of *opblue* after 2 h of light stimulation indicates that blue opsin may face faster protein turnover rates during light stimulation, which subsequently requires higher de novo protein production. Changes in expression for other opsin genes were non-significant. This may be due to the limited power to detect gene expression changes in our assay or because *opblue* functions as the main photoreceptor in this type of light entrainment. 

In mammals, light information from the photoreceptors is further processed by neurons in the retina, medulla, lobula, and other downstream neurons, including circadian clock neurons. Interestingly, in our study, one of the significant aspects of the enriched functional annotations of light-induced transcripts was related to cellular communication, especially the enrichment of glutamate metabolic processes, signal release, GABA receptor activity, neurotransmitter catabolic processes, and intracellular signal transduction. In particular, multiple types of *glutamate* and *nmda* receptors were found to be differentially expressed from the beginning of the light treatment and lasted until 2 h after it, implying light-activated glutamate receptor activities in neurons. Our results are in line with those obtained in *Drosophila*, where light input can be directly transmitted to clock neurons or can be indirectly transmitted through optic lobe interneurons [76]. In *Drosophila*, histamine is used as the main neurotransmitter by all photoreceptor cells, whereas photoreceptor cells in the Hofbauer–Buchner eyelets can signal to different neurons via an additional neurotransmitter—via histamine to the I-LN_v_ neurons or via acetylcholine to the s-LN_v_ neurons [27]. On the other hand, glutamate is the essential neurotransmitter in the mammalian circadian entrainment pathway. Light triggers the release of glutamate, activates N-methyl-D-aspartate (NMDA) receptors and nitric oxide (NO) signaling to phosphorylates, and activates the CREB/CRE pathway [77,78]. Additionally, GABA receptors are considered to be the primary neurotransmitters and can both synchronize and destabilize circadian rhythms in mammals, depending on the circadian phase or region in the SCN [79,80]. 

We also found support for a role of Ca^2+^ signaling in transferring the light information from the photoreceptors to the central clock. Action potentials from the release of glutamate are known to influence the NMDA-mediated currents and ion transportation—particularly Ca^2+^ transportation—in mammalian circadian phase shifts in the SCN neurons [81,82]. Therefore, it is interesting to find that gene transcripts related to annotations in calcium ion transmembrane transport (cluster 8) and inorganic ion transmembrane transport (cluster 5) were enriched in our study. In mammalian SCN neurons, the opening of the NMDA receptor led to the influx of Ca^2+^ into the cytoplasm, the phosphorylation of the CREB in the nucleus, activation of CREB/CRE binding, and, eventually, the induction of *per* clock gene expression [16,17]. Comparing our findings with the mammalian light input cascade, we found not only the significantly differential expression of three *creb* genes, but also the enrichment of potential CREB/CRE-binding motifs in clusters 3 and 10 of genes and, especially, in the upstream regions of two core clock genes, *cry2* and *npas2*. This activation of CREB/CRE pathway could potentially be the inducer of core clock gene expression in *Nasonia*’s light input pathway. 

At the core of the various processes that are affected by light exposure is the circadian clock, as indicated by the activation of the core clock genes *cry2* and *npas2* after 2 h of light stimulation. Although not statistically significant, the gene expression level of *cry2* appeared to be higher under light stimulation than in the dark control from the first time point (0.5 h). The expression of several other clock genes was not significantly affected by the light treatment, although this may have been due to the limited sample size and statistical power. The central clock system in *Nasonia* is thought to be comprised of CRY/PER and CLOCK/CYCLE (CLK/CYC), two types of heterodimers that form transcriptional–translational feedback loops. Although it is not well characterized in *Nasonia*, *npas2* is annotated as a paralog of the *clock* gene in mammals and in *Nasonia* [33]. Furthermore, *npas2* has overlapping roles with *clock* in circadian regulation in mammals [83]. In mammals, NPAS2 can also engage with BMAL1/BMAL2 to activate transcription of the *cry* and *per* clock genes, especially in peripheral tissues [84]. Therefore, it is possible that in *Nasonia*, light induces the expression level of *cry2*, which, in turn, facilitates the heterodimerization of CRY2 and PER and resets the circadian clock through a time-delayed transcription–translation negative-feedback loop. Additionally, light-triggered *npas2* gene expression may also contribute to the elevation of the *cry2* expression level after light stimulation. The light-induced upregulation of both *cry2* and *npas2* suggests that light mediates two points on the positive limb of the circadian transcriptional–translational feedback loop. This mechanism may reinforce the effect of light and, thus, result in a much stronger light-resetting response in *Nasonia* than in mammals, where this is only mediated by *per1/2* [39,85].

Light-induced *cry* gene expression has been well established in other vertebrates and some insects. In zebrafish, light drives the expression of both *cry1a* and *per2*, possibly inhibiting the clock through interactions with the PAS domain [23,86]. The FOS/AP-1 signaling pathway also mediates such a light-responsive pathway [87]. Additionally, several core clock genes, such as *Gb’cry1-2* and *Gb’per*, were upregulated in the newly proposed photic entrainment mechanism in crickets (*Gryllus bimaculatus*) [88,89]. Their research also proposed that light-induced *Gb’C-fosB* triggers the expression of *Fbxl* (*F-box and leucine rich repeat protein*) family genes, thus regulating TIM degradation in the crickets’ light-resetting pathway [88,90]. Even in mammals, the Ca^2+^-ERK1/2-AP-1 signaling pathway has recently been reported to regulate the light-induced *per* gene expression alongside the CREB/CRTC1-CRE pathway [91,92]. Therefore, it is possible that in *Nasonia*, *cry* functions as the core clock gene that is targeted by the light transduction cascade to reset the circadian oscillation. Additionally, the expression levels of the subunits of the *ap-1* transcription factor (composed of c-*fos* and *c-jun* heterodimers) were significantly affected by light after 2 h, with one being upregulated and one being downregulated. Combining the expression patterns of *ap-1* subunits and the enriched motif analysis, those transcription factors may bind to their motifs and modulate the gene expression of clusters 3, 4, and 9, resulting in the highest and lowest gene expression at 2 h. It is possible that the AP-1 signaling pathway controls *cry2* gene expression solely or in combination with the CREB/CRE pathway in *Nasonia*.

Overall, the central aim of our study was to characterize the light transduction cascade from the photoreceptor to the circadian machinery in *Nasonia*. The results suggest similarities involving light-induced clock gene expression in the circadian light input pathways in *Nasonia,* mammals, and other vertebrates. Based on the present findings and those of previous research [40], the most likely hypothesis of the light entrainment mechanism of the circadian clock in *Nasonia* is depicted in Figure 10. The process starts with light information that is processed by the opsin photoreceptor cells in the compound eyes and ocelli. Light information is further relayed to the clock neurons by means of neurotransmission, possibly via glutamate and/or GABA neurotransmitters and Ca^2+^ signaling. The combination of this neuronal signaling activates several transcriptional factors, including the CREB/CRE signaling pathway and the AP-1 signaling pathway, which subsequently promote the gene expression of *cry2* and *npas2*, indicating the two entry points of light information into the circadian transcriptional–translational feedback loop. *cry2* upregulation inhibits clock-controlled gene transcription, resetting the oscillatory mechanism. This proposed mechanism needs functional confirmation through behavioral experiments and knockdown/knockout approaches. As gene expression was measured in the whole head tissue containing brain and optic lobes, more detailed testing (single-cell sequencing or spatial cell sequencing) is needed to locate the light input pathway within specific clock neurons. In conclusion, our results shed light on the molecular mechanisms of the circadian light input pathway in hymenopterans and provide interesting candidates for a light-induced gene induction mechanism that resembles that of mammals and other vertebrates.

## 5. Conclusions

In this study, we aimed to characterize the light input transduction cascades in a hymenopteran insect, *Nasonia vitripennis*, to gain insight into circadian light entrainment pathways. The gene transcription levels in the head of *Nasonia* wasps showed pervasive changes during light stimulation, with 1886 genes being significantly differentially expressed during one or more time points from 0.5 to 4 h of light treatment. The genes included known clock genes, as well as many genes with functions in regulating gene transcription, translation, metabolism, and cell division. The strongest transcriptional response was recorded after 2 h of light treatment, with most genes being differentially expressed, including various genes implicated in the circadian clockwork. The joint results of co-expression analyses of all differentially expressed genes over the four time points, a motif analysis for transcription factor binding sites, and the differential expression of specific transcription factors indicate that especially CREB and AP-1 signaling were modulated by the light treatment. The transcriptional profiles for clock genes in *Nasonia* were strongly affected after receiving light stimulation in the middle of the subjective night. This may be an indicator that light induced the master regulator, which then triggered a cascade of changes in downstream genes. We proposed a possible signaling cascade from the opsin photoreceptor cells in the compound eyes and ocelli through the clock neurons via glutamate and/or GABA neurotransmitters and Ca^2+^ signaling to modulated CREB/CRE and AP-1 signaling, which subsequently promoted the expression of the two core clock genes, *cry2* and *npas2*.

## Figures and Tables

**Figure 1 biology-12-01215-f001:**
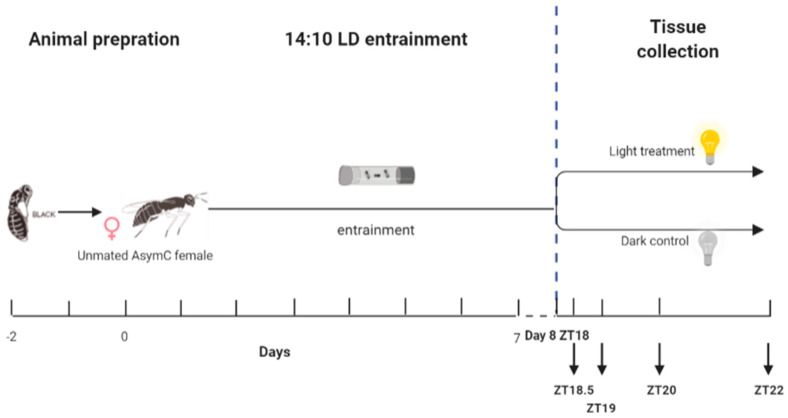
Setup of the RNA-seq experiment. One- to two-day-old unmated females were entrained under a 14:10 LD cycle for 7 days. On day 8, the LD cycle was switched off, and the wasps were kept in darkness. A light pulse was given starting on day 8 at ZT18 (i.e., 4 h into their subjective night and after 28 h of darkness), or wasps were left in constant darkness as a control. Samples were collected at 0.5 h (ZT18.5), 1 h (ZT19), 2 h (ZT20), and 4 h (ZT22) after the start of the treatment. The blue vertical dashed line indicates the onset of the treatment.

**Figure 2 biology-12-01215-f002:**
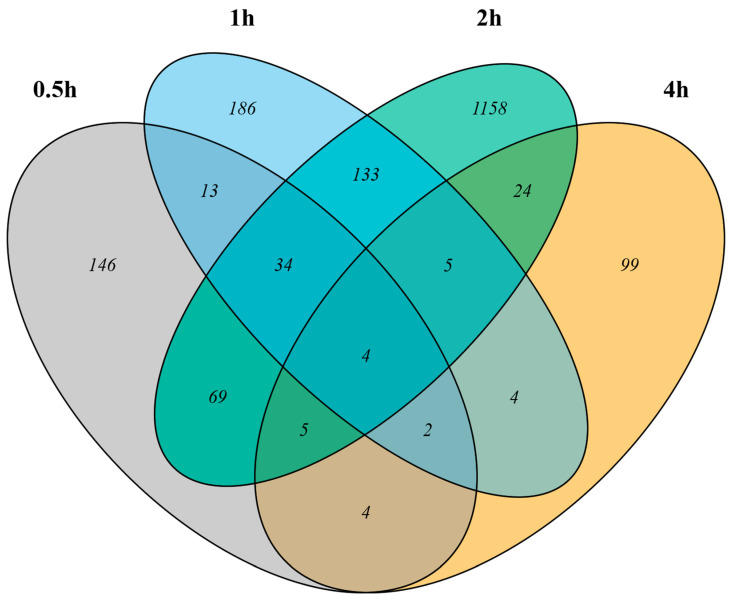
Light-induced changes in gene expression in wasp heads at four time points. Venn diagrams of the differentially expressed genes in response to light compared to the dark control at each time point. Of the 1886 genes that showed significant changes in expression in response to light, approximately 300 were differentially expressed at more than one of the four time points.

**Figure 3 biology-12-01215-f003:**
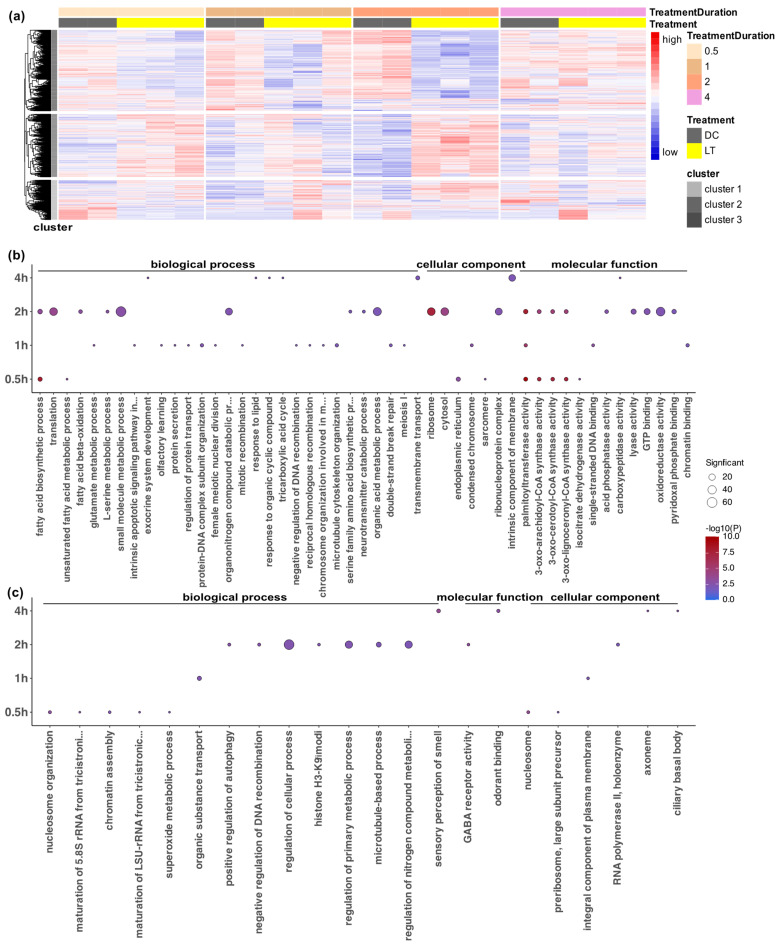
Overview of the global expression pattern and Gene Ontology (GO) enrichment analysis of the differentially expressed genes (DEGs) after the different durations of light stimulation. (**a**) Hierarchical clustering analysis of all 1886 DEGs showing their expression in all samples across all time points and treatments. The colored bars at the top indicate treatments and treatment durations. Regularized log (Rlog) data transformation and the z-score scaling method were used on raw count data. High gene expression levels are presented in red, and low expression levels are presented in blue. (**b**) GO analysis of upregulated DEGs after 0.5 h, 1 h, 2 h, and 4 h of light treatment compared to the dark control. (**c**) GO analysis of downregulated DEGs after 0.5 h, 1 h, 2 h, and 4 h of light exposure compared to the dark control. The *elim* algorithm and Fisher statistic test within the TopGO R package were applied for the gene overrepresentation analysis, and an adjusted *p*-value threshold of 0.01 was applied. The significance of the adjusted *p*-values is indicated by the color scale inside the dot plots. The size of the dots indicates the number of significant genes in that GO category.

**Figure 4 biology-12-01215-f004:**
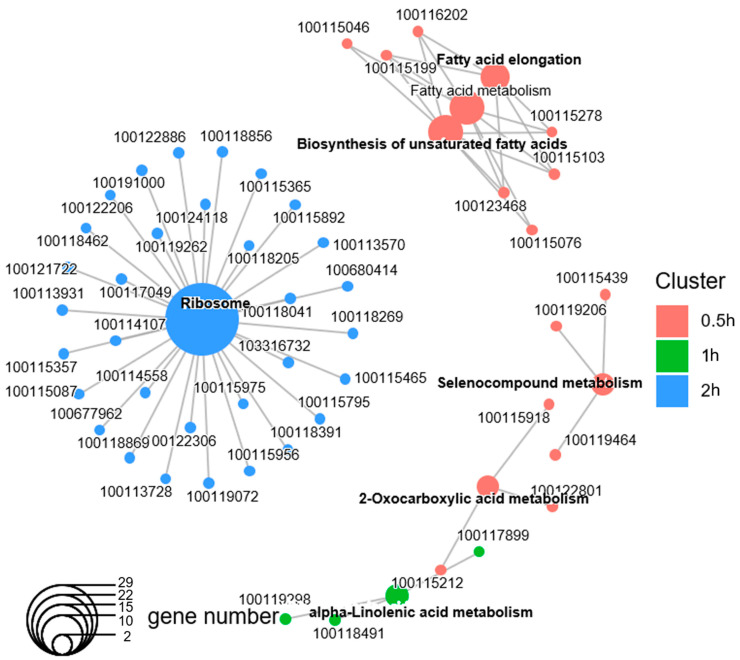
Kyoto Encyclopedia of Genes and Genomes (KEGGs) pathway enrichment analysis of the differentially expressed genes (DEGs) after each duration of light treatment. The gene–concept network displays which DEGs were involved in the specific significant KEGG pathway terms, which were clustered according to the DEGs from different time points. The colors indicate clusters from each time point, and no significant KEGG terms were found at 4 h.

**Figure 5 biology-12-01215-f005:**
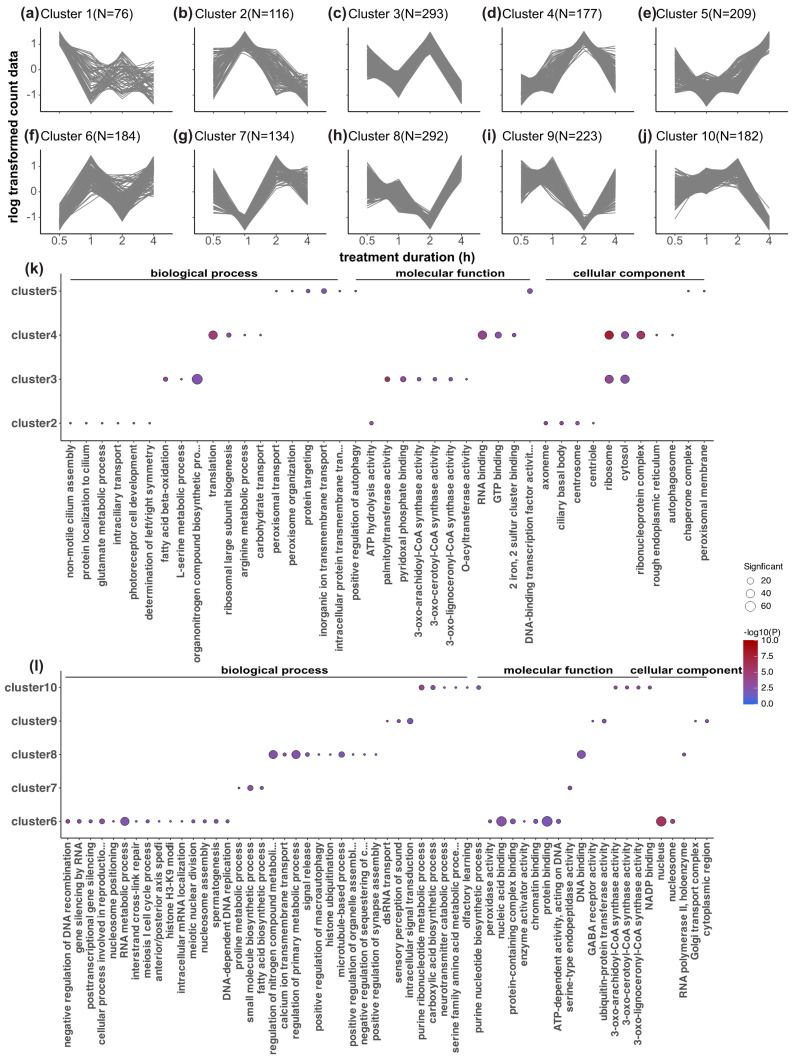
Time-course co-expression analysis of all DEGs and Gene Ontology (GO) enrichment analysis of each time-course cluster. (**a**–**j**) Ten different clusters of DEGs based on their co-expression patterns over time with the TCseq R package. Relative gene expression levels (expression differences between light-treated samples and the dark control) were used in the time-course clustering analysis to control for the changes in gene expression over time in the DC samples. Z-score scaling of the rlog-data-transformed relative expression is plotted in the line plots. The clusters reflect only the expression pattern over time but are not an indication of the significance of up- or downregulation. The number of DEGs in each cluster is denoted on top of each panel. The clusters for each DEG are listed in Appendix A. (**k**,**l**) Enriched GO terms for each cluster of DEGs from cluster 2 to cluster 10 (there are no significant GO terms for cluster 1). The *elim* algorithm and Fisher statistic test within the TopGO R package were applied for the gene overrepresentation analysis, and a strict adjusted *p*-value threshold of 0.01 was applied. The significance of the adjusted *p*-values is indicated by the color scale inside the heatmap plots. The size of the dots indicates a number of significant genes in that GO category.

**Figure 6 biology-12-01215-f006:**
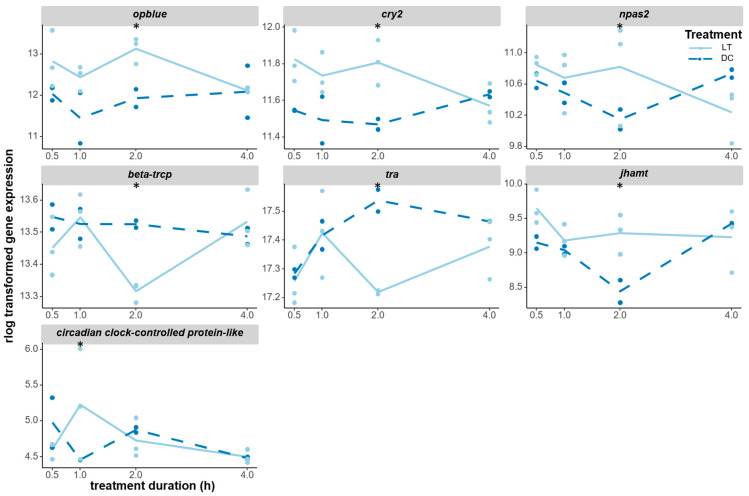
Expression changes of DEGs that were candidate genes for the circadian light input pathway. The *opblue* (*opsin, blue-sensitive*, LOC100121036) gene encodes for a blue-sensitive photopigment that is involved in circadian photoentrainment. *cry2* (*cryptochrome 2*, LOC100122802) and *npas2* (*neuronal pas domain-containing protein 2*, LOC100123168, a possible paralogous gene of the two *clock* genes) are two core clock genes that were triggered by light. *beta-trcp* (LOC100122997) is a possible homolog of the *slimb* gene in *Drosophila,* which is involved in the regulation of the circadian clock. *tra* (*transformer*, LOC100121203) is involved in sex determination, but also plays a role in photoperiodic regulation. *jhamt* (*juvenile hormone acid O-methyltransferase*, LOC100120870) is involved in the synthesis of juvenile hormone, which plays a role in diapause induction. *circadian-clock-controlled protein-like* (LOC100114918) is a potential clock output gene, but there is no specific information on it. The gene expression levels (raw counts) of these DEGs were normalized using rlog data transformation (statistical significance for upregulation or downregulation is indicated by an asterisk).

**Figure 7 biology-12-01215-f007:**
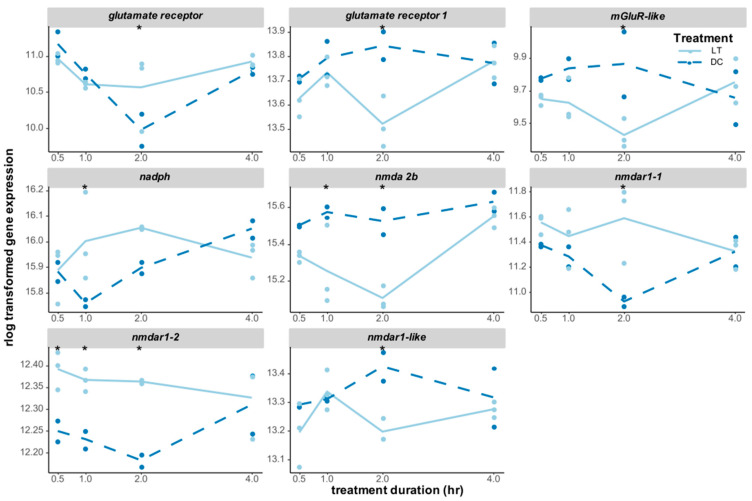
Expression changes of DEGs in the glutamate neurotransmitter pathway. *Glutamate receptor* (LOC100678247), *glutamate receptor 1* (LOC100116528), *mGluR-like* (*metabotropic glutamate receptor-like*, LOC100115971), *nadph* (*putative glutamate synthase [nadph]*, LOC100122109), *nmda 2b* (*glutamate receptor ionotropic, nmda 2b*, LOC100122774), *nmdar1-1* (*glutamate [nmda] receptor subunit 1*, LOC100116847), *nmdar1-2* (*glutamate [nmda] receptor subunit 1*, LOC116418325), and *nmdar1-like* (*glutamate [nmda] receptor subunit 1-like*, LOC103315526) were all significantly differentially expressed at least at one time point. The gene expression levels (raw counts) of these DEGs were normalized using rlog data transformation (statistical significance for upregulation or downregulation is indicated by an asterisk).

**Figure 8 biology-12-01215-f008:**
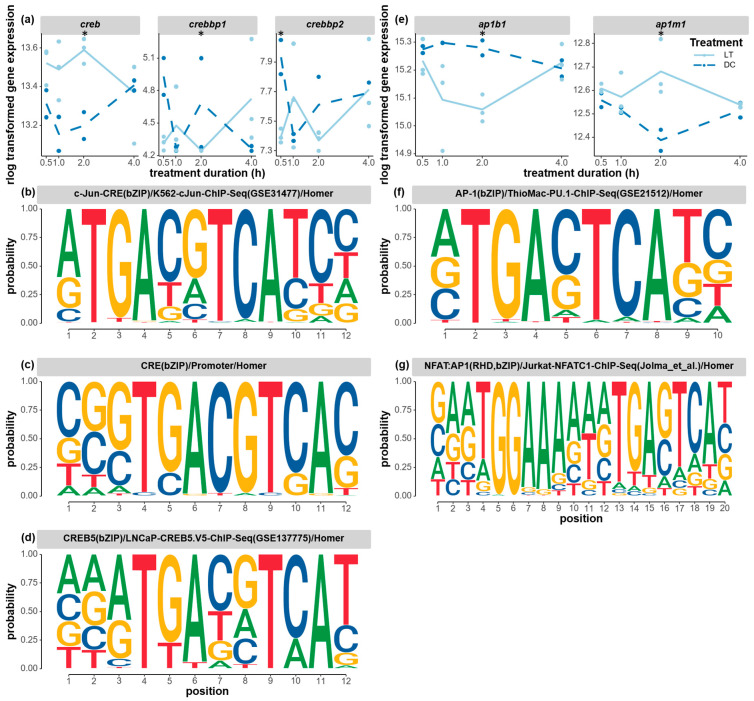
Expression changes of circadian transcription factors and the corresponding enriched binding motifs. (**a**) The gene expression levels of *creb* (*cyclic-amp response element binding protein*, LOC100121641), *crebbp1* (*creb-binding protein-like*, LOC103315501), and *crebbp2* (*creb-binding protein like*, LOC100679057) under the light and dark treatment. The enriched CRE/CREB-binding motifs found in the time-course cluster 3 (**b**), cluster 3/7 (**c**), and cluster 10 (**d**) genes. (**e**) The gene expression levels of *ap1b1* (*ap-1 complex subunit beta-1*, LOC100117777) and *ap1m1* (*ap-1 complex subunit mu-1*, LOC100114796) under the light and dark treatments. The enriched AP-1-binding motifs found in the time-course cluster 4/10 (**f**) and cluster 6/8 (**g**) genes. The gene expression levels (raw counts) of these DEGs were normalized using rlog data transformation (statistical significance for upregulation or downregulation is indicated by an asterisk). The Homer program was used to search for overrepresented known motifs in the upstream flanking region of the co-expressed cluster of genes. A *p*-value threshold of 0.01 was applied. The probability of each nucleotide at each position of the motif is indicated by the size of the nucleotide letter.

**Figure 9 biology-12-01215-f009:**
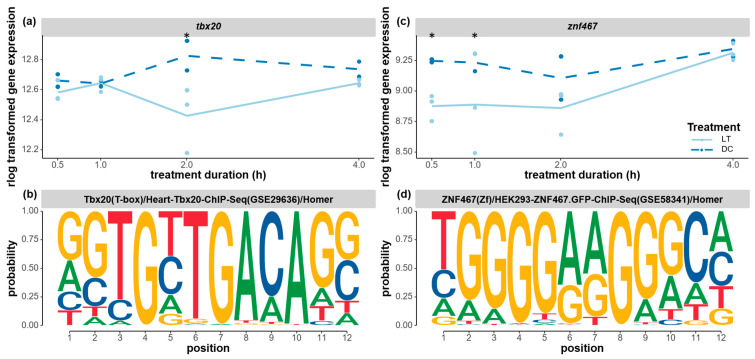
Expression changes of two developmental transcription factors and the corresponding enriched binding motifs. (**a**) The gene expression levels of *tbx20* (*t-box transcription factor tbx20-like*, LOC100117976) under the light and dark treatments. (**b**) The enriched TBX20-binding motif found in the time-course cluster 4 and 8 genes. (**c**) The gene expression levels of *znf467* (*zinc finger protein 467*, LOC100121593) under the light and dark treatments. (**d**) The enriched ZNF67-binding motif found in the time-course cluster 3 genes. The gene expression levels (raw counts) of these DEGs were normalized using rlog data transformation (statistical significance for upregulation or downregulation is indicated by an asterisk). The Homer program was used to search for known overrepresented motifs in the upstream flanking region of the co-expressed cluster of genes. A *p*-value threshold of 0.01 was applied. The probability of each nucleotide at each position of the motif was indicated by the size of the nucleotide letter.

**Figure 10 biology-12-01215-f010:**
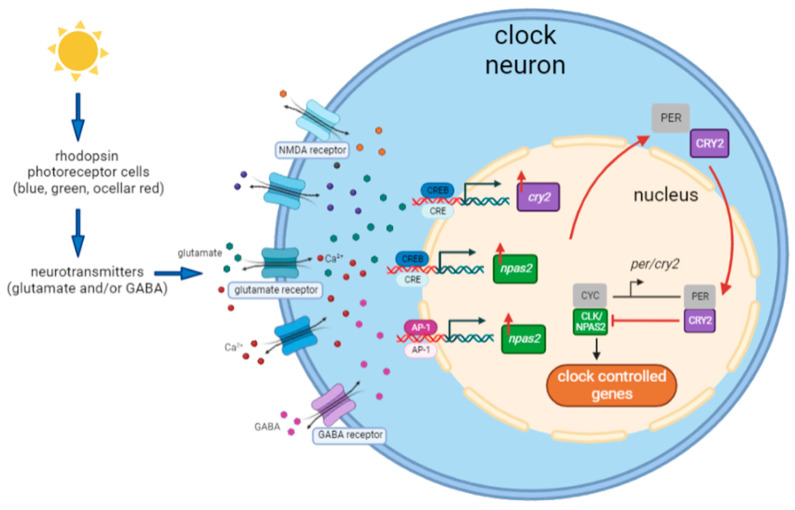
A model for the light entrainment mechanism of the circadian clock of *Nasonia*. Light is perceived by the rhodopsin photoreceptor cells (including blue, green, and ocellary red photoreceptor classes). The information is transmitted to clock neurons through neurotransmitters—possibly glutamate and/or GABA. This neurotransmission regulates glutamate receptors, NMDA receptors, and/or GABA receptors and potentially cause the influx of glutamate, GABA, and Ca^2+^. This neuronal signaling then activates the CREB/CRE signaling pathway and AP-1 signaling pathway, leading to the upregulation of *cry2* and *npas2* gene expression. *npas2* may also contribute to the upregulation of *cry2* expression. Eventually, upregulation of *cry2* results in CRY2/PER heterodimerization and resets the clock by inhibiting CLK-/CYC-controlled gene expression.

## Data Availability

The raw RNAseq reads used in this article can be found in the European Nucleotide Archive (ENA) under accession no. PRJEB57723. The processed data used for analysis, including the gene count matrix, sample information, GO annotation table, and Appendix A, can be found on Dryad (https://doi.org/10.5061/dryad.jq2bvq8dq, published on 3 September 2023). Bioinformatic and statistical analysis scripts that can be used to reproduce the results can be found on GitHub at https://github.com/YFWang-YvH/Transcriptomic-analysis-of-light-induced-genes-in-Nasonia-vitripennis-implications-for-circadian (published on 3 September 2023).

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
