# Peer review of "Transcriptomic Analysis of Light-Induced Genes in Nasonia vitripennis: Possible Implications for Circadian Light Entrainment Pathways"

_biology, 2023, doi:10.3390/biology12091215_

Round 1

Reviewer 1 Report

The manuscript by Wang and Hut et al focuses on characterizing the light-induced transcriptome changes in Nasonia vitripennis using RNAseq and bioinformatic methods. The manuscript is well-written with a clear experimental setup and an adequate bioinformatic pipeline. This reviewer has some comments that aim to stimulate the authors to investigate this exciting dataset further and provide a clear message for a general audience.

Overall, I consider that this manuscript has the potential to be accepted in Biology journal after a major revision.

Main comments:

1.              Although the authors mention and discuss the effect of time, the experimental setup does not account for any circadian aspect. The experimental model accounts only for an acute response in response to light. Using the same sample number, the authors could have performed a profile of 3 samples every 6h. This would allow the identification of rhythmic aspects rather than the acute light effects. Therefore, I suggest the authors focus on the acute light aspect of the introduction, including the title, instead of the circadian aspect as the manuscript does not focus on the circadian aspect, but rather on acute light effects.

2.              A major limitation of the DEGs analysis is the uneven sample number and an n of 2 in the control group. The authors mention this in the discussion, but I recommend that the authors reinforce this important limitation further.

3.              The authors provide important information regarding the biological processes of each DEG class in figure 3. However, the figure layout is very poor and difficult to read it. The authors must provide a novel way to plot these findings. I suggest the authors choose the top-5 processes for each category and plot them. In addition to adding, the number of genes, the authors should also provide the p-value for each enriched process.

4.              In Figure 3, the authors describe the most affected pathways in response to light. However, it is elusive what is the final consequence of the animal’s physiology. Could the authors predict such outcomes based on the RNAseq data?

a.     It would be interesting the authors to provide information on common and exclusive pathways. This could be done by using upset plots.

b.     Using different enrichment tools often brings conflicting results due to the marked differences between the tools. The authors must discuss the pathways that are highlighted in different methods.  

5.              Figure 5 needs to be improved (see the above comment). The labeling is confusing and must be fixed. In figure 5, panel A and B (mid-row) are duplicated. Please correct this mistake.

6.              What is the main message of figure 5? The authors describe very nicely the different clusters of DEGs and their respective processes. However, it is elusive what is the physiological relevance.

a.     One suggestion would be to evaluate how many similar or the same biological process are enriched in different clusters. How many are unique and shared?

b.     The authors should provide a straightforward message to the reader as in this current version, the reader is overwhelmed with data.

7.              How similar are the motif regions between humans and wasps? The authors focus only on 500 bp regions. However, promoter regions can be present up to 10,000 and still affect gene expression. I found it complex to use human motif data to make proper comparisons. The authors should provide a clear justification to include human data as I consider that only Drosophila database should be included in the manuscript.

Minor comments

1.     The authors should provide a PCA plot as an introductory figure.

2.     Could the authors explain why only female wasps were used?

3.     Since only female wasps were used, the authors must discuss gender effects on the manuscript.

4.     How collection in the darkness was performed. Please provide this information.

5.     None of the data availability datasets are available. Please correct this.

6.     The table S2 file needs to be correct. Remove “stat” termination to allow proper file opening.

7.     In table S3, p value is not provided. This information is required.

Only minor spelling and grammar mistakes were identified. 

Reviewer 2 Report

This is a very interesting article. Authors have performed a transcriptomic analysis comparing gene expression after light pulses of different frequency during the darkness, to identify putative novel light entrainment pathways in hymenopteran. The manuscript is very well written and structured, methods are well described, it seems to be well performed and it has a very interesting and deep discussion suggesting putative light entrainment pathway for hymenopteran.

However, I have few minor comments. During all article authors demonstrated to have a profound knowledge on light circadian system entrainment in Drosophila and mammals (mainly from results obtained only in rodents in fact), but they do not mention other vertebrates. There are other light entrainment systems, especially in fish, in which it is well known that cry1a and per2a are directly light-induced. It could be very interesting to look for important genes for putative pathways of light entrainment, of not only mammals but fishes, as these vertebrates are completely absent in this MS.

Some good reviews could be found (see below) but there are much more.

Steindal and Whitmore. 2020. Zebrafish Circadian Clock Entrainment and the Importance of Broad Spectral Light Sensitivity. Front Physiol. 14;11:1002. doi:10.3389/fphys.2020.01002.

Pagano et al. 2018. Evolution shapes the responsiveness of the D-box enhancer element to light and reactive oxygen species in vertebrates. Sci Rep. 4; 8:13180. doi:10.1038/s41598-018-31570-8.

Isorna et al. 2017. Interplay between the endocrine and circadian systems in fishes. J Endocrinol. 232:R141-R159. doi: 10.1530/JOE-16-0330.

Uchida et al. 2010. A common origin: signaling similarities in the regulation of the circadian clock and DNA damage responses. Biol Pharm Bull. 33:535-44. doi:10.1248/bpb.33.535.

Vatine et al. 2011. It's time to swim! Zebrafish and the circadian clock.FEBS Lett. 585:1485-94. doi: 10.1016/j.febslet.2011.04.00.

I also have some other minor specific comments:

-        Lines 36-37, please include a reference.

-        Line 43, I think that the word “circadian” before photoreceptors should be deleted.

-        Line 45-35. This is a too long paragraph, it is necessary all this information on Drosophila and mammals? As mentioned fish are not mentioned here. Instead, maybe information regarding development of photosensitivity in Nasonia vitripennis could be very interesting in order to understand why developmental stages have been selected.

-        Maybe line 122 and line 114 have same information? It is better explained in line 122.

-        Lines 133-137 is maybe a part of discussion but I think it should not be included in materials and methods section.

-        Explain what is considered darkness for sampling (lines 147). Complete darkness is not possible, what kind (colour, intensity) of light was employed?

-        How was determine significant differences? (line 215), FDR<0.05 with which test?

-        Lines 368-371, it seems that this paragraph is contradictory, how do you explain?

-        It could be possible to transform the data in rlog units in something related with fold change?

-        Line 426-427, how can we localize the 63 transcription factors in table S3? Such factors are up or down regulated?

Reviewer 3 Report

The manuscript by Wang et al. reports their gene expression study in the parasitic wasp Nasonia. They studied the transcriptional response in the head following a light pulse during the night, which reset the circadian clock system. This study is particularly interesting because, unlike Drosophila, the canonical insect model in chronobiology, Nasonia (like all other hymenopteran insects) lacks the dedicated circadian photoreceptor CRY1. The study reveals interesting genes and pathways that are involved in light entrainment and could serve as the first steppingstone to molecular dissection of this system. This study suggests that compared to Drosophila, the system in Nasonia shows a higher resemblance to the mammalian system. Overall, this is an interesting study with valuable data that would be interesting for the readership of Biology journal.

However, there are a few major problems that need to be addressed. First, the experimental design is problematic because the two factors are confounded: time after the pulse start and pulse duration. Therefore, the increase in the number of differentially expressed genes (DEGs) during the experiment could be a result of either accumulation of transcripts that are slower to respond to the onset of the stimulus, or a larger number of transcripts that respond to a longer duration of the stimulus. This problem should at least be acknowledged in the manuscript.

Second, the way by which genes are defined as DEGs is problematic. I don't think that running 4 separate tests, each for a different time point, and having at least one significant test, is an adequate statistical procedure. Instead, you should use an ANOVA design that considers all 4-time points at the 2 conditions simultaneously. This would allow you to test the interaction between treatments. The authors' definition of DEGs leads to problems in the interpretation of their following analysis, for example, the heatmap in Figure 3. They make generalizations of transcript behavior at different time points, although these specific time points were not necessarily the ones that were significantly different between the conditions. The same goes for the GO analysis, where specific time points are tested, but not necessarily different in given transcripts.

The third problem is apparent when the authors present specific candidate genes. One can see that the expression level of the control samples (that should remain constant across all samples) deviates at specific time points, and this local deviation leads to a significant statistical difference between the conditions. Thus, many of the DEGs seem to be false positives due to noise in the control. The authors need to figure out a way to better filter this noise and to use a more stringent method for the identification of DEGs.

Minor comments

Line 56 Please explain what's PACAP.

Line 64: I don't think that this distinction between two light resetting mechanisms that are supposedly represented in Drosophila and

mammals is real. There is a fast transcription response to light in Drosophila as well.

Line 119, please explain why you used females (as opposed to males).

Line 139. what was the light pulse's duration and intensity?

Line 142, this is less than ideal experimental design, because two factors are confounded: time after the pulse start and pulse duration.

Line 158. Please explain the vertical blue line.

line 176, please state the amount of total RNA used for cDNA synthesis.

Line 209, please explain what type of data normalization was carried out before gene expression comparison, and if no normalization was used, please justify why.

Line 213, the experimental design is less than ideal, as there are two confounding factors: the time passed from stimulus onset and the stimulus duration.

Line 215: define FDR here.

Line 215: I don't think that running 4 separate tests, each for a different time point, and having at least one significant test, is adequate statistical procedure. You should use an ANOVA design that considers all 4-time points at the  2 conditions simultaneously. Possibly testing for interaction between treatments

Line 215: how do you treat genes wherein two or more groups there was a differential expression, but in opposite directions?

Line 216, Please state what the type of distance used for clustering and what kind of clustering was used (e.g. single-linkage,

average-linkage)

line 241, replace the spaces in the numbers with commas.

Line 251. Please move the definition of FDR to the first use of the term (line 215)

Line 253, because two independent variables were confounded (see above), you don't really know whether expression "increased with duration of light treatment", or simply reflected the gradual accumulation of transcripts responding to the light onset (from early to late response)

line 256: how do you consider genes wherein two (or more) time points, the expression changed significantly but in opposite directions?

line 273: "using gene set over-representation" is not clear. Please reword.

line 273: were these p-value adjusted for multiple testing? they should

Line 275: For most genes, in 3 out of 4-time points, there was no significant change. It makes sense to submit only the subset of genes for each time point, for which there was a significant difference.

Line 301, Figure 3a. The heatmap indicates a large difference between the 3 biological replicates of duration 1 hr, treatment=light. In the first cluster two replicates show low expression and the one to the right is high. In the second cluster, this is reversed. two replicates are high expression and the third one low. How is that possible? this suggests a technical problem in the analysis.

line 318: again, the so-called DEG differ from the control only at a single time point. So the temporal pattern in Figure 5 does not necessarily represent a significant difference from control.

line 328: in my opinion, this is the only GO analysis that should be presented. The earlier analysis should be omitted from the manuscript.

Line 338, "indicating a light input pathway starting from...." sentence is not clear, please revise.

Line 355: Y-axis label of the bottom histogram should be near the axis, the Y-axis number should be shown near each graph (or at least next to (a).

Line 365, 0.01 is not a "strict p-value". were these p-values adjusted for multiple testing, as they should? if yes, state how, if not, please do, or justify.

Line 377 Figure 6. One expects to see a uniform expression of the control (DC). However, in the case of tra and "circadian clock-controlled protein-like", the expression of the control deviates from flat line. It is the deviation of the control that results in these genes being considered DEG. In the case of circadian clock-controlled protein-like", the large variation of the 3 replicates suggests that this is a false positive.

Line 405: Fig. 7. the same problem as in figure 6. the difference in expression is a result of deviation of expression of the control which is obviously an artefact. This is apparent for "glutamate receptor", nadph, nmdar1-1 and ndar1-like. all these examples should be therefore omitted.

line 433 insert "of the" before "identified". change to "clusters". remove "respectively.

Line 465: figure 8. Same problem as above. in creb, ap1b1, ap1m1, one can see a trough in the control, that contributes to the differential expression. this seems to be false DEG.

Line 610. A recent study in cricket also reported up-regulation of cry 2 after a light pulse. you may want to cite this reference: https://doi.org/10.3390/ijms231911358

Line 648, I don't think the data suggest "strong resemblance". Please tone down this statement.

There are a few style problems (see minor comments).  I have only pointed out a few of them. Please consider the use of a proofreading service.

Round 2

Reviewer 1 Report

The authors have addressed all my criticism and I consider this paper suitable for publication.